# An open-source, high-resolution, automated fluorescence microscope

**Ando Christian Zehrer[1], Ana Martin-Villalba[2], Benedict Diederich[3]\*, Helge Ewers[1]\***

[1]Institut für Chemie und Biochemie, Freie Universität Berlin, Berlin, Germany; [2]Department of Molecular Neurobiology, German Cancer Research Cente, Heidelberg, Germany; [3]Leibniz-IPHT Jena, Jena, Germany

**Abstract** Fluorescence microscopy is a fundamental tool in the life sciences, but the availability of sophisticated equipment required to yield high-quality, quantitative data is a major bottleneck in data production in many laboratories worldwide. This problem has long been recognized and the abundancy of low-cost electronics and the simplification of fabrication through 3D-printing have led to the emergence of open-source scientific hardware as a research field. Cost effective fluorescence microscopes can be assembled from cheaply mass-produced components, but lag behind commercial solutions in image quality. On the other hand, blueprints of sophisticated microscopes such as light-sheet or super-resolution systems, custom-assembled from high quality parts, are available, but require a high level of expertise from the user. Here, we combine the UC2 microscopy toolbox with high-quality components and integrated electronics and software to assemble an automated high-resolution fluorescence microscope. Using this microscope, we demonstrate high resolution fluorescence imaging for fixed and live samples. When operated inside an incubator, long-term live-cell imaging over several days was possible. Our microscope reaches single molecule sensitivity, and we performed single particle tracking and SMLM super-resolution microscopy experiments in cells. Our setup costs a fraction of its commercially available counterparts but still provides a maximum of capabilities and image quality. We thus provide a proof of concept that high quality scientific data can be generated by lay users with a low-budget system and open-source software. Our system can be used for routine imaging in laboratories that do not have the means to acquire commercial systems and through its affordability can serve as teaching material to students.

**\*For correspondence:** benedict.diederich@leibniz-ipht. de (BD); helge.ewers@fu-berlin.de (HE)

## eLife assessment

This **important** study provides **compelling** evidence that the low-cost and open-hardware UC2 microscopy framework can be expanded to enable single-molecule localization microscopy. The authors managed to fit the instrumentation and control thereof in a unit that can be placed in a small stage-top-incubator. Together with providing adapted software for data acquisition and data analysis, the UC.STORM setup can rival the capabilities of comparable commercial instruments at a fraction of the costs.

## Introduction

Fluorescence microscopy is, due to its excellent contrast, specificity and compatibility with live cell imaging, an essential tool in the life sciences. Digital imaging, quantitative analysis and ever higher throughput and resolution in new techniques make fluorescence microscopy more important than ever. Nevertheless, while the development of high-end instrumentation pushes forward our progress in understanding cellular processes, spatial and temporal resolution are not the only challenges impeding progress in scientific discovery. This is because necessarily, the access to cutting-edge

instrumentation is reserved to a subset of scientists as a result of its cost and technical complexity, effectively limiting data production to laboratories with access to substantial resources. At the same time, cost of electronic or optical components, powerful processors and high-quality cameras are at an all-time low. Together with simple prototyping through 3D-printing, this has led to the emergence of open-source scientific hardware as a research field (*Wenzel, 2023*). The open-source microscopy community has experienced the rapid development of different approaches aiming to make microscopy more available (*Almada et al., 2019*; *Alsamsam et al., 2022*; *Ambrose et al., 2020*; *Auer et al., 2018*; *Danial et al., 2022*; *Diederich et al., 2019a*; *Hohlbein et al., 2022*; *Sharkey et al., 2016*; *Voigt et al., 2019*; *Wenzel, 2023*).

The first iterations of open-source microscopes came from microscopy laboratories that tried to provide similar quality data as commercial systems while being assembled from components for lower overall cost; however, still requiring significant expertise. Solutions now exist for single molecule localization microscopy (SMLM; *Alsamsam et al., 2022*; *Auer et al., 2018*; *Holm et al., 2014*; *Kwakwa et al., 2016*; *Martens et al., 2019*), they include modular platforms (*Li et al., 2020a*), light sources (*Berry et al., 2021*; *Schröder et al., 2020*) and even sophisticated special solutions such as light sheet microscopy (*Voigt et al., 2019*), spinning disc microscopes (*Halpern et al., 2022*), and high-throughput applications (*Li et al., 2019*; *Walzik et al., 2015*).

On the other hand, another group of investigators tried to provide extremely cost-effective microscopy solutions that are accessible to everyone. The result are solutions that can be useful in detecting for example parasites in inaccessible regions of the world for simple diagnosis (*Li et al., 2019*), but are incompatible with quantitative scientific imaging (*Maia Chagas et al., 2017*; *Cybulski et al., 2014*; *Diederich et al., 2020*; *Diederich et al., 2019b*; *Sharkey et al., 2016*; *Hernández Vera et al., 2016*). Most of these solutions lack excitation and emission for fluorescence imaging.

Here, we propose an instrument filling the gap between the low-cost open-source instruments for the general public and high-quality fluorescence imaging, as required for scientific research in molecular cell biology. We provide an automated high-resolution microscope with single molecule sensitivity that can be fully assembled and steered by lay users. Our system is based on the UC2 toolbox (*Diederich et al., 2019a*; *Diederich et al., 2020*; *Wang et al., 2022*) using common optical elements, affordable electronic components and 3D printed parts and is combined with a high NA objective, motorized stage and intuitive GUI-based operation.

The result is a cost efficient and user-friendly high-resolution fluorescence microscope that is compatible with long-term live-cell imaging, single particle tracking and single molecule localization microscopy. All designs for the 3D parts, software and electronics are open-source (*Table 1*). We here describe microscope assembly, benchmarking and a number of high-resolution microscopy assays performed with our microscope.

## Results
### Assembly of a high-resolution UC2 microscope
We aimed to generate a cost-effective, high-resolution widefield fluorescence microscope that allows us to perform a variety of common cell biological imaging experiments. These include standard high-resolution diffraction limited immunofluorescence imaging, long-term live-cell imaging, single particle tracking and super-resolution microscopy. To fulfill these criteria, we reasoned that our system should

**Table 1.** Github repositories for hardware, electronics and software.

| Repository | Link: |
|---|---|
| General openUC2 | https://github.com/openUC2/UC2-GIT; copy archived at *Diederich, 2024* |
| Specific to this setup: UC.STORM | https://github.com/openUC2/UC2-STORM-and-Fluorescence, (*openUC2, 2024b*) |
| ImSwitch (UC2 specific) | https://github.com/openUC2/ImSwitch/. (*openUC2, 2024c*) |
| UC2-Rest | https://github.com/openUC2/UC2-REST, (*openUC2, 2024a*) |
| UC2-ESP32 | https://github.com/youseetoo/uc2-esp32, (*youseetoo, 2024*) |
| UC2-ESP32 Firmware Flashintool (Online) | https://youseetoo.github.io/ |

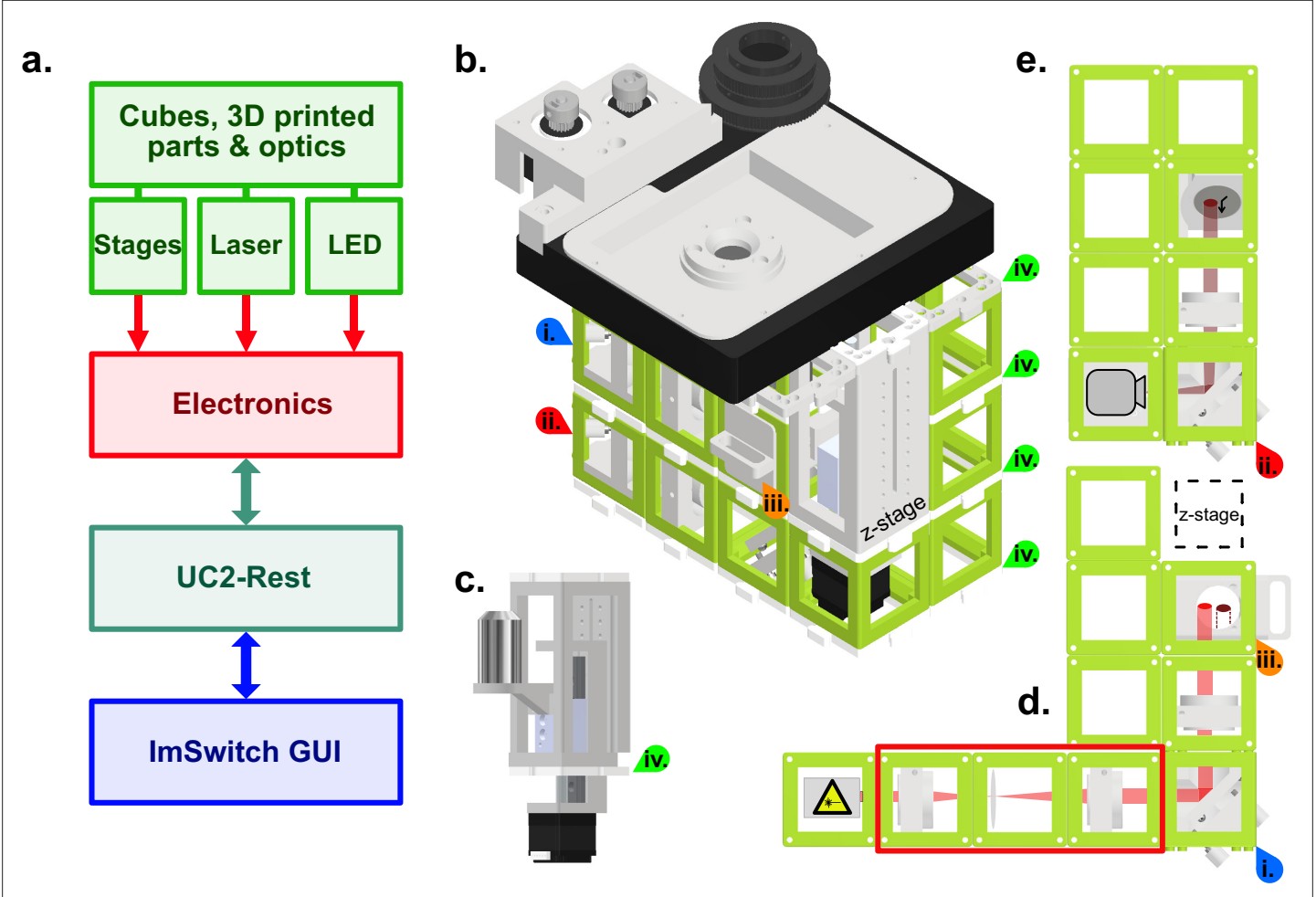

**Figure 1.** Schematic of the high-resolution-UC2 widefield microscope. (**a**) Overview of the different component categories of the microscope, from the hardware, through the electronics controlling, to the software and the ImSwitch-based GUI. (**b**) Schematic of the complete setup. Red and blue arrows (**i** and **ii**) are references to help visualize the 3D structure. Top layer is the commercially bought XY-stage that was motorized for this project. Timing belts convert motor torque (two grey gears top left) into stage motion (bigger black gear middle) but have been omitted in this representation for clarity. Sample holder can be printed according to the sample used, circular in this case. (**c**) Commercially bought precision Z-stage and high-NA objective (Olympus 60 x/1.49 NA TIRF) (**d**) Excitation layer. Laser emission (bright red light-path) is focused by a *f*=100mm lens onto the back focal plane of the objective. Orange arrow (**iii**) marks the filter cube which has an excitation filter (Chroma ZET635/20 x EX), a dichroic mirror (Chroma ZT640rdc) and on the bottom an emission filter (Chroma ET655lp long-pass for $\lambda$ >655 nm) to separate excitation from fluorescence. Red rectangle shows a telescope build including a diffusor (rotating piece of cling foil) in the focus point between both lenses. Telescope build magnifies and homogenizes the laser beam and can be used optionally. (**e**) Detection layer, corresponds to the bottom layer in the setup (red star as reference). Emitted fluorescence (dark red light-path) is depicted with a *f*=100 mm lens on the detector (Alvium 1800 U-158c CMOS camera from Allied Vision). All active optical elements, for example mirrors and lenses are mounted into the injection moulded cubes (50x50 × 50mm, green) via custom 3D printed mounts. Cubes are connected laterally and vertically via the puzzle shaped connection pieces. Green arrows (**iv**) indicate the connecting puzzle layers between the cubes.

have: (i) excitation and detection powerful and sensitive enough for single molecule localization, thus requiring a laser and an appropriate detector. (ii) capability to operate inside an incubator for long-term live-cell imaging, thus requiring small size and automation. (iii) an intuitive and user-friendly GUI for maximal accessibility. Lastly, the entire system including the physical microscope, the electronics and the software must be open-source and be possible to assemble and operate with moderate technical know-how (*Figure 1a*). We reasoned that the quality of the objective is paramount to image quality, while most other optical parts of a light microscope can be mass-produced components. Furthermore, such a high-quality objective allows for the swift diagnosis of problems in the assembly of the optical path due to its high resolution and minimal chromatic and spatial aberrations. We decided to use an Olympus 1.49 NA oil objective, but any high-quality objective can be incorporated in our system, by adapting the objective mount to the requirements of the objectives' manufacturer.

Based on the guidelines described above and to ensure adaptability and modularity, we decided to build our open-source automated microscope on the basis of the UC2 toolbox using its injection molded cubes for the structural assembly. For ease of usage, we used an inverted configuration and assembled the microscope in three layers to keep the size and design compact (*Figure 1b*). The high-precision x-y sample table and z-stage (*Figure 1c*) were low-cost commercial solutions that we motorized and implemented into the open-source, python-based microscope GUI as automated elements of the microscope. The emission pathway at the bottom of the microscope assembly is based on an Alvium 1800 U-158c CMOS camera from Allied Vision, which compared to conventional scientific cameras is more affordable by a factor of 30–60 times at the time of submission (*Figure 1d*), more compact and consumes less energy. One important point here is that the camera needs to be monochromatic as the Bayer pattern on a polychromatic camera would significantly reduce the sampling as well as the quantum efficiency of a camera chip, which is already lower compared to that of scientific cameras. The camera was connected to the computer via USB 3.0, allowing fast data transfer and

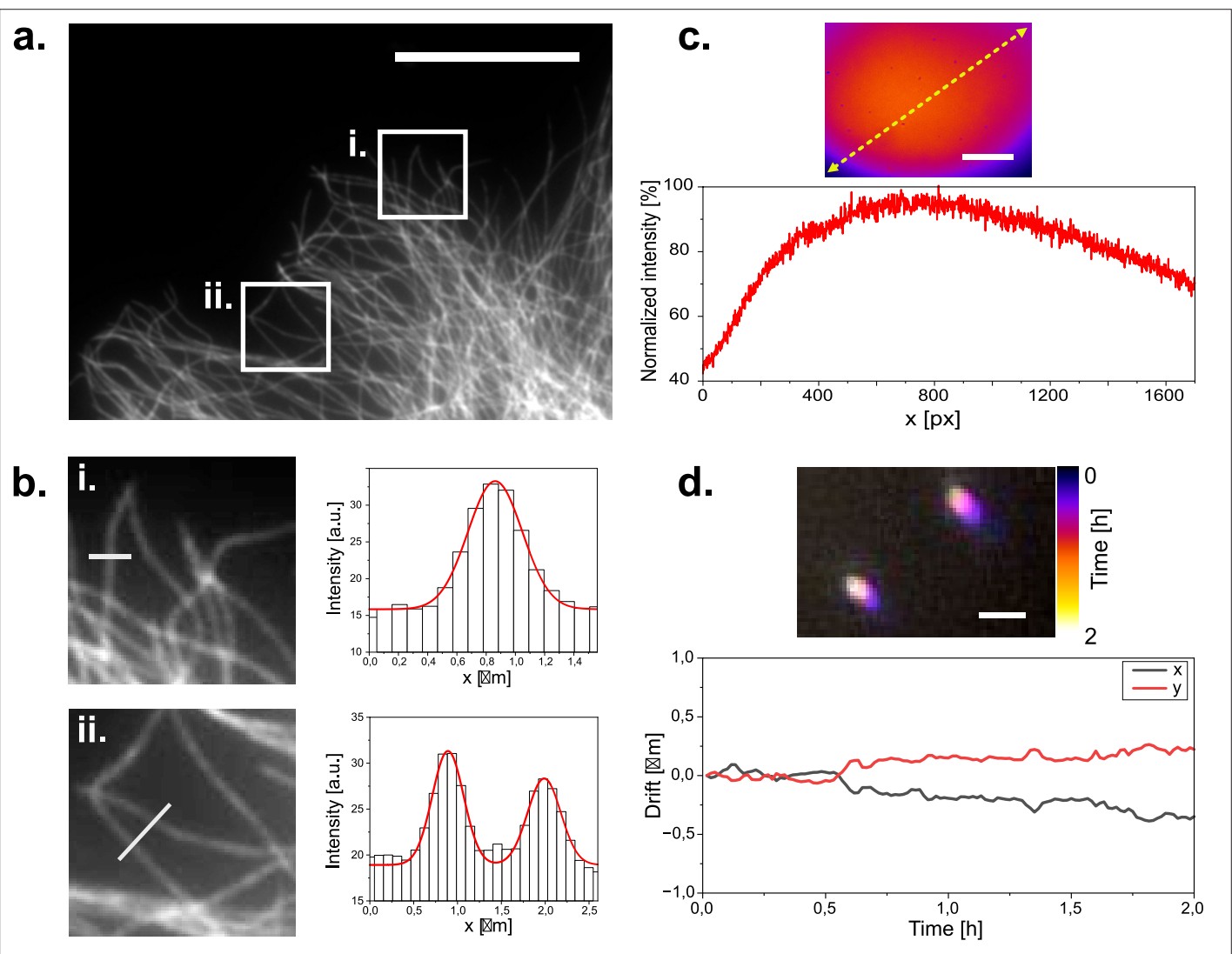

**Figure 2.** Widefield imaging of immuno-stained CV1 cells against tubulin (α and β-tubulin). (**a**) Widefield image of a tubulin stained CV1 cell. Two regions of interest are highlighted. Scale bar represents 20 µm. (**b**) Zoomed in images of the selected ROIs in (**a**). with respective microtubule profile plotted across the white line in the ROI images. ROI dimensions are 8x8 µm. Grey values are plotted as columns and fitted with gaussian functions (red line). Microtubule diameter can be estimated to 340–420 nm according to the full width at half maximum of the fits. (**c**) Illumination profile of whole camera chip (1456x1088 px) at the sample plane as imaged on a far-red fluorescent Chroma slide. Scale bar represents 35 µm. (**d**) Tetra-speck beads imaged over a period of three hours and color coded over time. Scale bar represents 1 µm. Bead positions are localized over time to measure their displacement to their initial position over time.

maximal compatibility. Between the emission pathway and the sample stage, the excitation pathway contained a low-cost 638 nm diode laser with an optional beam expander (*Figure 1e*), that depending on the illumination requirements could be inserted or removed, modulating the illumination density and homogeneity of the excitation field (see *Figure 2c*). To trigger the laser, control laser power and to steer the x, y and z motors for focus and sample translation, we used a low-cost Arduino-based board that can be connected via USB3.0 to any computer. Electronics on the board and the USB-camera are then directly controlled via an adapted GUI of the python-based open-source software ImSwitch (*Moreno et al., 2021*), resulting in a fully automated microscope. Additionally, ImSwitch includes the 'ImScript' an automation feature, which allows to execute a sequence of written *Python* commands with user defined parameters for example overnight time-series imaging experiments. Furthermore, we took advantage of the open-source (OS) nature of ImSwitch and implemented the recently presented OS SMLM algorithms microEye (*Alsamsam et al., 2022*) to realize an online-running localization framework that gives instant qualitative feedback if the sample preparation works as expected. In addition, the feedback of these online-running image processing algorithms can be looped back into the hardware control to eventually compensate possible drift (e.g. autofocus). A large variety of available Napari (*Damm et al., 2023*) plugins can be used to directly work on the data without moving them to a different computer.

All steerable electronics could alternatively be controlled with a PS4 controller for increased user-friendliness. The assembly can be housed with 3D printed walls or even black cardboard to block ambient light and increase laser safety. In *Table 1*, a list of the different microscope and software elements and the repository addresses to the corresponding Materials and methods is displayed. A comprehensive online repository provides the bill of materials (BOM), as well as a set of instructions on how to setup the soft- and hardware (*Table 1*).

The 638 nm Red Laser Module used for illumination induced a recurrent pattern in the detection path when imaging a coverslip homogeneously covered with fluorescent molecules. Spectrometric measurement of the excitation laser beam confirmed that our low-cost laser had minor bands in the wavelength range of 650–670 nm, which we could remove using an additional band pass excitation filter (ZET635/20 X). Our finding encouraged us in our approach to mix low-budget components with high quality optics. Focusing and positioning of the sample is done with commercial low-budget stages that offer great long-term stability. To render all three axes of movement computer controlled, we first motorized an x-y stage via torque transmission form the motor to the micrometer screws of the stage via timing belts. Secondly, we incorporated a motorized lead screw-driven (NEMA11, 50 mm, Amazon, 50€) z-stage holding the objective into the modular design of the microscope.

## Imaging experiments

We first used our microscope to image a sample of fixed mammalian cells stained against tubulin via immunofluorescence using AF647. When we imaged these cells with moderate laser intensity (35 W/cm$^2$), beam expander and 50ms integration time, we found that cells were evenly illuminated (*Figure 2a*) and microtubules appeared as diffraction-limited lines inside cells (*Figure 2b*). To quantify flatness of illumination, we made use of a red auto-fluorescent benchmarking slide (Chroma). The measured fluorescence intensity was quite homogenous across the field of view, with variations of up to 25% of the maximal fluorescence intensity, excluding the bottom left and right corners (*Figure 2c*).

These conditions are compatible with high-quality fluorescence microscopy in life science research. To test the stability of our system, we then imaged sub-diffraction fluorescent beads repeatedly over time. When we then calculated the displacement of the beads over time, we consistently found drift of less than 1 µm over 2 hr (*Figure 2d*). We concluded that our microscope may be suitable for imaging experiments that require repeated measurements of the same field of view such as live-cell imaging. We thus mounted live cells at room temperature on the optical table on our microscope and stained them with SiR-actin (Spirochrome). When we then repeatedly illuminated the cells with moderate laser power (18 W/cm$^2$) to excite SiR-actin fluorescence and with white light in transmission mode to image the cells, we found that the cells tolerated acquisition conditions for several hours without damage visible in fluorescence or widefield image (*Figure 3*). As expected, drift was negligible. Importantly, we found similar stability against drift over time, when we performed this experiment not on an optical table, but a wetlab bench over a layer of foam material (made of Polyethylene or alternatively Poly-urethane) as a damping layer. This suggested that our microscope might be suitable for long-term

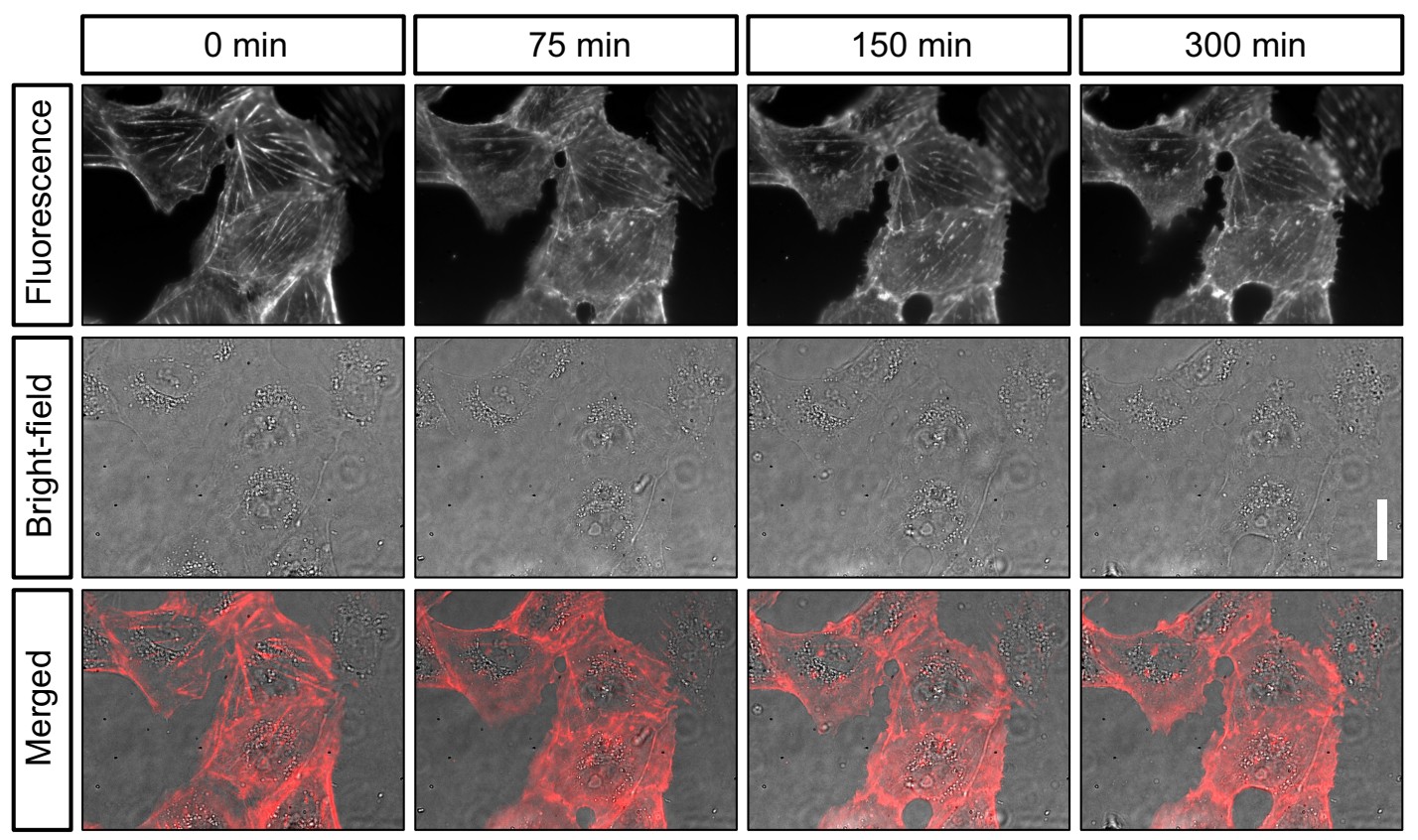

**Figure 3.** Live-cell imaging of actin in cultured cells. Shown are still images of the same CV-1 fibroblast cells in SiR-actin fluorescence and widefield (background corrected) at different timepoints over 5 hr. Scalebar on the right represents 30 μm.

imaging in an environment that does not provide vibration damping, such as a cell culture incubator. Given that our microscope assembly is of small size, and connected via two flexible USB cables, we then moved on to perform live-cell imaging over longer periods in a cell culture incubator.

To do so, we used a standard 37 °C $CO_2$ incubator. The longer USB cables connecting the microscope to the exterior allowed for steering from the outside. When we plated live T98G cancer cells on coverslips and mounted them on the microscope sample holder in warm medium inside the incubator, they exhibited normal morphology and growth over many hours. When we performed a long-term imaging experiment using Syto deep-red fluorescence staining of the nucleus (Thermo Fisher), we found that cells kept dividing and moving on the coverslip for several days (*Figure 4*). We concluded that our setup was compatible with long-term live-cell imaging of cells.

Next, we aimed to further explore the single molecule capabilities of our microscope and performed single particle tracking (SPT) experiments on live CV-1 African green monkey cells stably expressing GPI-GFP. When we incubated these cells with streptavidin 655 nm quantum dots (QDs) coupled to biotin-tagged anti-GFP nanobodies (*Figure 5a*), we detected QDs readily as single diffraction limited entities on cells (*Figure 5b*). When we then localized emission centers and tracked the QDs over time, we could generate trajectories over many frames (*Figure 5c and d*) and extract parameters describing particle movement such as the diffusion coefficient (D, *Figure 5e*) and the moment scaling spectrum (MSS, *Figure 5f*; *Ewers et al., 2005*). The MSS introduces a parameter that allows to quantitatively describe the type of motion a particle undergoes (MSS = 0 being immobility,<0.5 Brownian diffusion and >0.5 super-diffusive). Our measurements were in agreement with our previous observations (*Li et al., 2020b*).

Since we were readily able to detect single quantum dots, which are single emitters of few nanometers in size, we reasoned, it might be possible to perform single molecule localization microscopy (SMLM) experiments as well. We immuno-stained CV-1 cells against tubulin using AF647 coupled secondary antibodies and incubated coverslips in BME/GLOX buffer for effective pumping into the

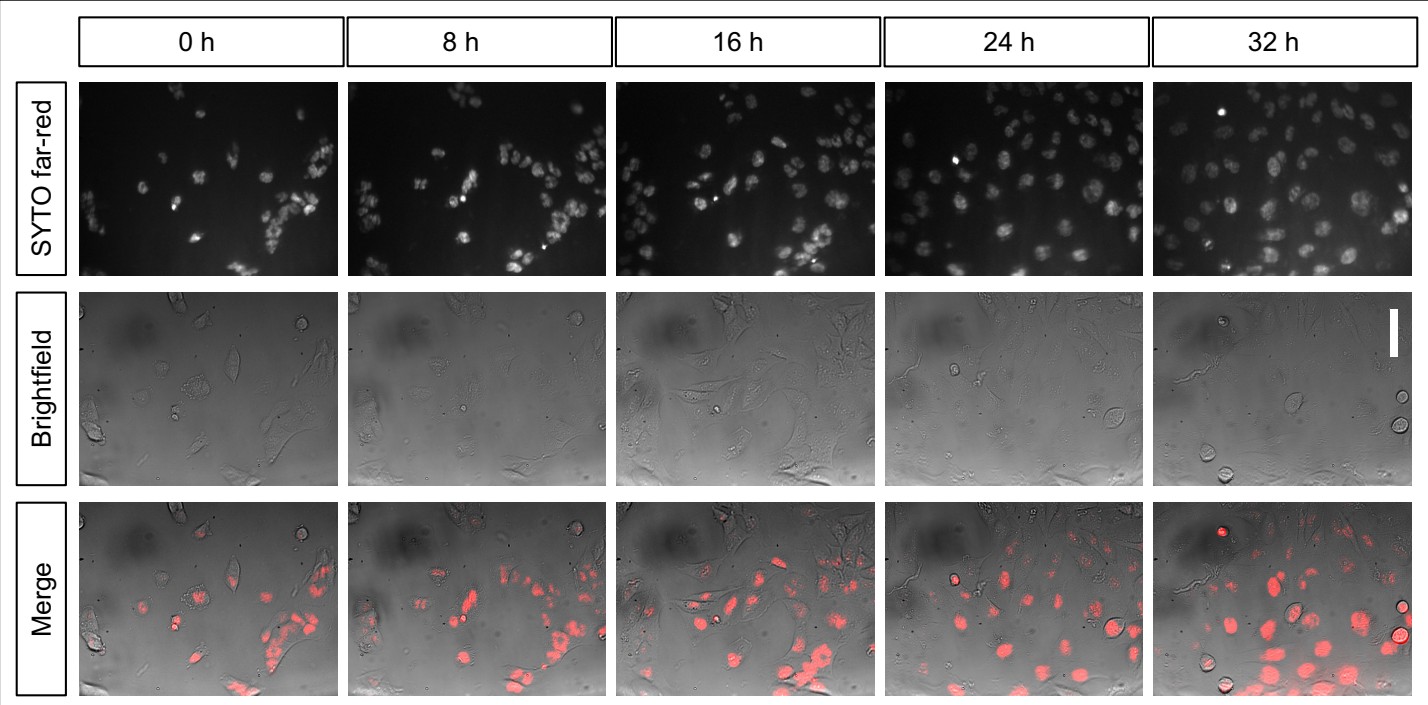

**Figure 4.** Long-term live-cell imaging of cultured cells inside the incubator. Shown are still images of the same population of T98G glioblastoma cells over time in the same field of view. Cells are stained with SYTO nucleic-acid fluorescent dye and imaged in widefield (background corrected) every 20 min for over 32 hr inside an incubator. Scalebar on the right represents 30 µm.

dark state. When we then illuminated our sample with high intensity laser light (in our case ~1kW/cm$^2$), we found that the intense staining quickly faded and gave way to the typically observed blinking in SMLM experiments. We took a timeseries of 30,000 frames and localized the detected emitters using *Thunderstorm* (*Ovesný et al., 2014*). Even though we were limited to widefield illumination, we achieved a drastic improvement in resolution in reconstructed SMLM micrographs (*Figure 6a*). When we measured the profile of individual microtubules, we could easily detect the typical 'railroad track' pattern with a spacing of 38–43 nm, indicative of high-quality SMLM imaging (*Figure 6b and c*). To quantify the achieved resolution, we used Fourier ring correlation (*Nieuwenhuizen et al., 2013*) and found that the overall achieved resolution was 93 nm, clearly below the optical resolution limit (*Figure 6d*).

## Discussion

Here, we developed a cost effective, high-quality automated microscope that can be assembled by laypeople with minimal background knowledge in optics, electronics and informatics. The assembly guidelines including CAD files of the 3D printed parts, list of used components and software are openly accessible on the Github repositories (see *Table 1*), thus offering maximal availability.

Our microscope is based on a simple modular assembly system called UC2 (You.See.Too., *Diederich et al., 2019a*). This framework is based on modular cubes that can hold different optical components in place and allow for simplified alignment and assembly. This flexibility furthermore facilitates the replacement of any lens, filter, laser camera or other components, allowing the user to implement already available parts into their setup, or to further trim down the setup by removing unused parts. We made use of an improved version where the cubes are made through injection molding, which provides a significantly higher manufacturing precision than the same cubes fabricated using 3D-printing, resulting in greater mechanical stability and reproducibility, while being available at low cost through mass production (*Wang et al., 2022*). The hereby provided increased structural integrity makes the setup portable allows it to be reproducibly built and dismantled within hours. This facilitates its use in remote areas. Improved versions or updated parts could easily be shared digitally and

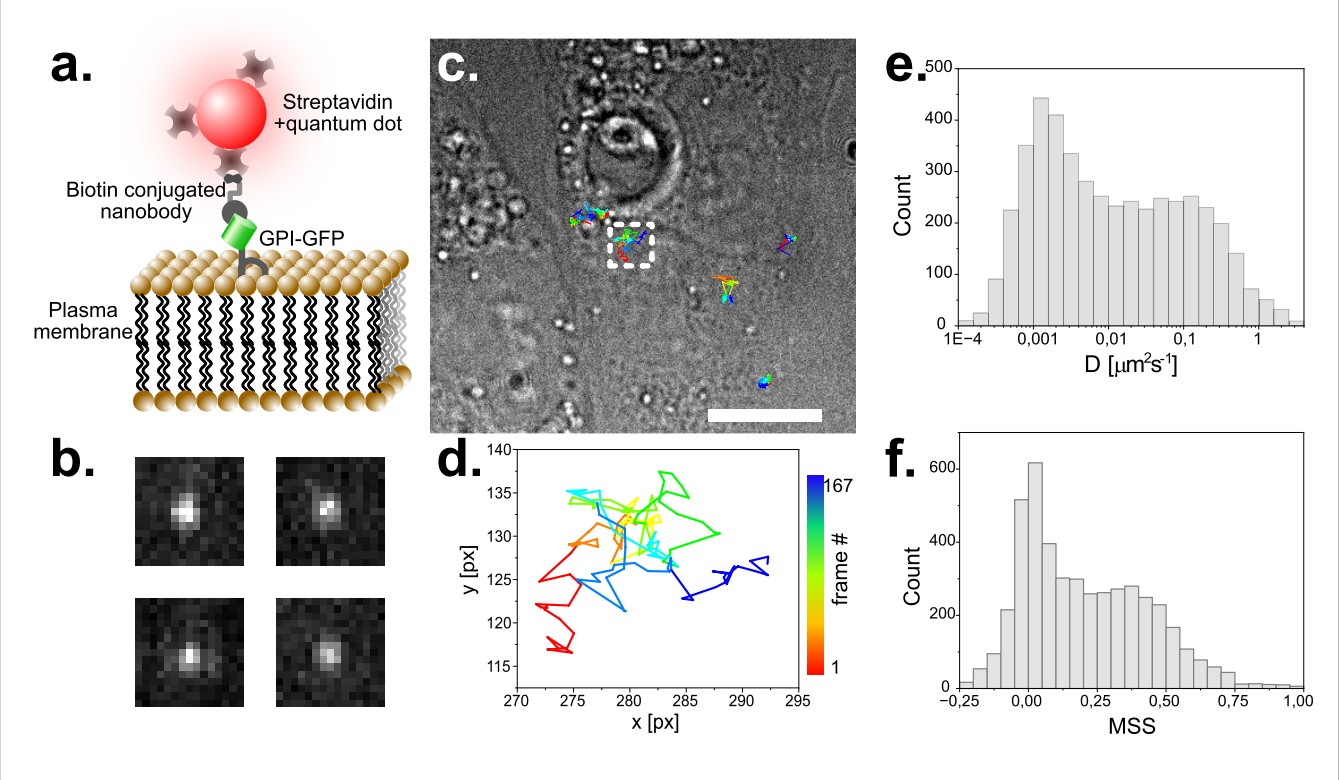

**Figure 5.** Single particle tracking of GPI-GFP in live CV1 cells using functionalized Quantum dots. (**a**) Schematic of the construct used to track single GPI- molecules with quantum dots. (**b**) Exemplary raw images of single fluorescent Quantum Dots. Image dimension is 1.6 μm x 1.6 μm. (**c**) Trajectories of several single molecules visualized on a top-light illuminated image of CV1 cells. Scale bar represents 10 μm. (**d**) Selected trajectory of a single particle over 167 time-points. Sampling rate is 25ms, pixel size 104 nm. (**e**) Histogram of the diffusion coefficient of around 5000 tracks with 20 or more detected time points each. (**f**) MSS (moment scaling spectrum) of the same population as plotted in (**e**).

printed locally, further promoting the advantage of digital manufacturing in the laboratory. Here, the created assembly can easily be replicated multiple times to increase imaging throughput still for a fraction of the cost of a single high-end microscope.

We present data in standard immunofluorescence microscopy, live-cell microscopy and long-term live-cell microscopy assays performed inside an incubator. Furthermore, we show single particle tracking data of semiconductor quantum dots on live cells and single molecule localization based super-resolution microscopy. In all cases, the quality of data we generated is comparable to those generated by systems orders of magnitude more expensive.

Our system thus allows for sophisticated imaging assays to be performed at a professional level with greatly reduced cost. At the same time, electronic control of components and remote steering of the microscope can be implemented without background knowledge in electronics. This makes the generation of high-quality image data accessible especially for life science research groups with limited monetary means also in lower income countries. The main limitation of our system in this direction is the requirement of a high-NA objective, the by far most expensive component of the setup. Since for scientific imaging, high quality data and thus high-NA and corrected objectives are essential, this component cannot be replaced. However, the quality of low-cost objectives has recently been improving, even with objectives allowing for effective single molecule detection. The reduced NA diminishes the localization precision and the signal to noise ratio, thus limiting the possible applications. Development of brighter dyes might also be an alternative to improve the imaging quality enough to account for low budget lasers and filters.

Due to its modular assembly, extensions may be easily implemented to our microscope. Depending on the specific research aim, multiple excitation and emission wavelengths may be desirable, but would require a redesign of the frame to incorporate motorized filter switching and more light sources. Alternatively, multiband filters in combination with fiber-coupled and thus

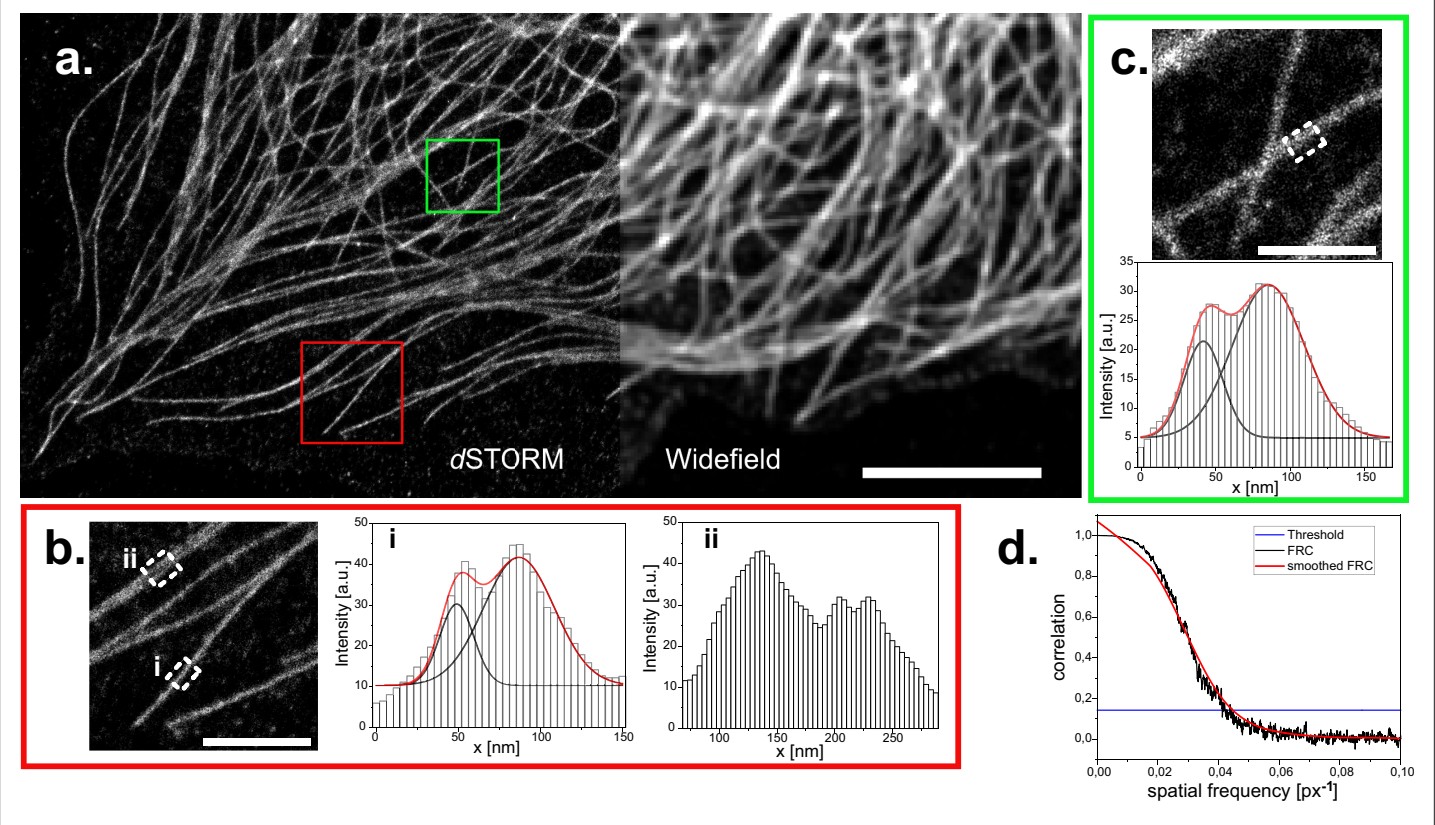

**Figure 6.** dSTORM reconstruction of immuno-stained CV1 cells against tubulin (α and β-tubulin). (**a**) Widefield image and reconstructed super-resolution image of 30'000 widefield frames, localized and reconstructed using ThunderSTORM, applied parameters described in the methods part. Reconstruction is displayed with the average shifted histogram method, including a 25-fold magnification. Scale bar represents 5 µm. Two regions of interest in the reconstruction are marked (red square for (**b**) and green square for (**c**)) and zoomed in (scale bar 1 µm). Profiles within these ROIs are measured by averaging the profile of the reconstructed image over 250 nm along the microtubule (white dashed rectangles). White histogram bars are the intensity values of the pixels in the reconstruction. Two peaks arise from the circular shape of microtubules. These were analyzed with a double gaussian fit (red curves are the cumulated single black fits). Distances of (38±2) nm (red box) and (43±2) nm (green box) could be extrapolated. (**d**) Fourier ring correlation of the complete reconstructed image. Threshold is left to the preset 1/7. Overall resolution is estimated at 93 nm.

combinable laser sources are also possible. In the future, the possibilities of this system could be significantly expanded by adding more modules specific for additional capabilities and we specifically invite the community to do so. Especially, the combination with a low-budget spinning disc (*Halpern et al., 2022*), TIRF illumination or light-sheet imaging (*Diederich et al., 2020*) are possible future applications.

Further steps towards user-friendliness include the replacement of the laser light-source by a high-power LED to reduce the requirement of laser safety. Another step would be to render the electronics assembly fully plugable, thus eliminating the soldering steps that are required in the current assembly.

Besides the application in basic research, this cheap and easy to assemble setup would be useful for teaching the next generation of scientists that more than ever have to be able to synthesize capabilities spanning several disciplines. We endorse making scientific instruments available for more scientists, students, as well as the general public. Our system builds on efforts in open science regarding software (ImSwitch, *Moreno et al., 2021*) and microscope hardware (UC2 *Diederich et al., 2019a*) while benefitting from a growing field of open-source microscopy and laboratory hardware in general (*Almada et al., 2019*; *Auer et al., 2018*; *Hohlbein et al., 2022*; *Martens et al., 2019*; *Wenzel, 2023*). Taken together with efforts in making fluorescent dyes available (*Lavis, 2021*), fundamental obstacles to the free diffusion of capabilities and know-how in science are being removed. We hope that our live-cell high-resolution incubator microscope (U.C.STORM) will catalyze these efforts.

## Methods

### Component fabrication

In the components list, a detailed table describes all the different pieces used for the microscope, including the price of each component. The CAD designs as well as further information about UC2 or setup building can be found on the Github repository (see *Table 1*). The XY-stage was a manual commercial model that was modified and motorized for the application. The Z stage was also commercially bought, but already manufactured with a motor. Both stages were chosen because of their low price and convenient use. Custom-made stage solutions to print and build can be found on the UC2 repository (*Table 1*). In the slicing software Prusa slicer, the original Prusa i3 MK3S & MK3S+printer was set. The parts were printed with the 0.15mm SPEED or 0.15 mm QUALITY settings. The two materials used in this project were the Prusament PLA (Polylactic Acid) and Prusament ABS (Acrylnitril-Butadien-Styrol) filaments (both from Prusa), which can both be selected in the slicing software ABS was used wherever heavy loads may deform the setup in the event of temperature gradients (e.g. objective mount). Materials were printed with PLA infill (20–40%) ($T_{print}$ = 215 °C; $T_{bed}$ = 60 °C). Materials printed with ABS infill (20–40%) ($T_{print}$ = 255 °C; $T_{bed}$ = 100 °C).

### UC2 setup assembly

In the UC2 system, optical components were mounted in standardized cubes. To make alignment possible over an optical excitation or emission path over several cubes distance, the components must experience a large enough degree of freedom within the cubes so they can be moved manually in angle and exact position, but low enough to keep components immobile over prolonged time periods. The challenge presented by tilted optics can on the one side be addressed during the design process; broader mounts/adapters are less prone to tilt due to higher contact surface with the cube. Once the optics were orthogonal to the light path, their position in relation to the optical axis was set manually with highest possible accuracy, using appropriate mounts. The fine adjustment was done with tiltable mirrors, as using translational mounts for every optical element would be complex to produce or costly. A helpful tool for adjustment was to position an aperture in the excitation pathway. The maximum of the laser spot was set to be in the center of the aperture hole and was adjusted using the mirror. The emission path was adapted in the same way, as imaging a fluorescent surface allows to position the maximal laser intensity spot on the center of the detector field of view (also with a mirror).

The setup entails two imaging channels: the laser induced fluorescence (excitation path) and the LED top-light illumination (650 nm Star LED, Cree, 3 W, brightfield). The top-light light source is mechanically detangled from the stages to remain unaltered by sample motion. As the filter cube in the microscope is immobile, the top-light illumination requires either a high-power white LED or an LED that emits light in a wavelength range that is not blocked by the dichromatic mirror or the emission filter (in our case 650–660 nm).

### Setup parts availability and price

The different 3D printed parts, electronics and optics used in the assembly, the corresponding CAD files as well as an estimation of the price can be found in the corresponding Github repositories, as listed in *Table 1*. General information about the UC2 system can be found on the UC2 repository.

### Image acquisition and control software

For the microscopy control, acquisition and reconstruction software, we chose ImSwitch because its modular design adapts very well to the inherently modular nature of the cube-based optics platform. A novel firmware "UC2-ESP32" was developed, which is fully integrated into the python-based ImSwitch structure. Using a unified REST API-like interface, lasers, motors, LEDs, LED arrays and readout sensors can be controlled serially via WiFi and USB. To ensure wide distribution of this device adapter, we used a readily available hardware module that can control multiple stepper motors and easily adapt other hardware components such as the LEDs and lasers. An online-based flash tool loads the latest firmware onto the Wemos ESP32 D1 R32 in combination with the CNC Shield to create a fully functional microscopy control unit. Detailed documentation can be found in the online documentation (https://youseetoo.github.io/). A Python library 'UC2-REST' adapts the USB serial and integrates it into the ImSwitch software. In this project, the strengths of open-source were fully unleashed by extending the control and acquisition algorithm ImSwitch with an online SMLM reconstruction

algorithm by *Alsamsam et al., 2022*. The user can select different parameters from the MicroEye framework to locate blinking events directly in the Napari viewer (*Sofroniew et al., 2022*) to directly track possible reconstruction or sample artefacts. The UC2-specific fork amongst other repositories can be found in *Table 1*.

## Imaging modalities

### Illumination

The illumination is provided by the 638 nm Red Laser Module 500 mW Round Dot Focusable TTL 3050 (acquired on https://www.laserlands.net/). When implementing the telescope and beam homogenizer, the illumination density reaches up to 90 $W/cm^2$ but could be tuned to any required value below that. In the focal point between both lenses a rotating piece of cling film diffuses the beam. The variance in the laser density over the field of view excluding the corners was measured with a fluorescent slide and is reproducibly lower than 25%.

Additionally, for long term experiments, lacking fluorescence intensity can to an extent be compensated with longer detector exposure times. To perform SMLM that is *d*STORM in our case, the telescope is removed to have maximal laser density. The total available power at the sample plane is 83 mW thus a laser density of around 520$W/cm^2$. But since the laser spot is smaller than the FOV, sufficient laser density can be reached when only using a subset of the FOV (approximately a third of the FOV) can be used with sufficient laser power (over 970$W/cm^2$) and a variation in laser intensity of about 25% of the laser intensity maximum.

### Objectives and pixel sizes

For the widefield imaging, the room temperature live cell imaging, the single particle tracking as well as the *d*STORM experiment, a 60 x Olympus 1.49NA oil immersion objective (UPLAPO60XOHR) is used. The setup optics have accordingly been chosen for maximal resolution thus matching the Nyquist sampling rate on the camera. The pixel-size is determined with a calibrated grid and a value of 104.1 nm is extrapolated. For the incubator measurements, a 20 x Olympus 0.6NA air (MXPLFLN20X) is used. The optics in the setup are not adjusted to this objective, resulting in a pixel-size of 319.4 nm.

## Coverslip and cell seeding

### Cell lines

CV-1 cells (Wild-type Cercopithecus aethiops kidney fibroblasts, ATCC CCL-70, mycoplasm free, verified), HeLa (human cervical canerepithelial cell line ATCC CCL-2, mycoplasm free, verified) and T98G (human glioblastoma multiforme cell, line ATCC CRL-1690, mycoplasm free, verified).

### Coverslip preparation

Metal holders were used in combination with circular coverslips (25 mm) which were first cleaned with a 2% Hellmanex solution for 10 min then cleaned in ethanol for 10 min and were finally placed in the plasma cleaner for 15 min after having dried. The eight-well sample-holders (Ibidi) were washed with 1 M KOH for 10 min then rinsed with $H_2O$ and dried.

### Cell culture medium

Cell culture medium was on a phenol red free DMEM base with 1% GlutaMAX and 10% FBS.

### Cell seeding

Cells were detached from the cell culture dish with a 2 mM EDTA solution. A total of 30,000 cells were seeded on a 25 mm circular coverslip for *d*STORM experiments and general wide-field imaging. For the live cell experiments, 55,000–80,000 cells were seeded onto 25 mm coverslips, alternatively 15,000–25,000 cells per well when using an eight-well chamber.

## Microtubule staining

CV-1 cells were first washed with 37 °C warm PEM buffer. The cells were fixed with warm PEM with 4% PFA, 0.05% GA and 0.1% TX100 for 20 min. After fixation, sample was quenched in 8 min with $NH_4Cl/PBS$ (50 mM) then washed three times with PEM buffer. The cells were then permeabilized with

0.3 % TX100 in PEM for 5 min and washed with PEM buffer. The sample was then blocked for 30 min in imageIT followed by 1 hr in a 4% HS, 1% BSA, 0.1 % TX100 PEM-based solution.

The sample was incubated overnight (12 hr) at 4 °C with a primary mouse anti alpha- and beta-tubulin IgG antibodies (Sigma-Aldrich (REF: T5168) and Sigma-Aldrich (REF: T5293)) diluted by a factor of 1–350 in a 1% BSA and 0.1 % TX100 PEM based solution. Sample was thoroughly washed three times for 5 min with a 1% BSA and 0.1 % TX100 PEM-based solution. The cells were then incubated with the secondary donkey anti-mouse antibody conjugated with the AF647 fluorophore, diluted by a factor of 1–250 in a 1% BSA and 0.1 % TX100 PEM based solution at room temperature for 2 hr. The three washing steps were then repeated, as previously mentioned for the primary antibody.

The sample was the post-fixated with a 4% PFA diluted in PEM solution, for 15 min. The cells were then quenched with a 50 mM $NH_4Cl$ solution diluted in PBS for 8 min. Finally, the cells was washed 3 times.

## Single particle tracking

Anti-GFP nanobodies (LaG-16 anti-GFP, own production according to *Fabricius et al., 2018*) were added to a biotin (α-Biotin-$\omega$-(succinimidyl propionate)–24(ethylene glycol)) solution at double molar excess. After 1 hr of incubation on a shaker (300 rpm), the nanobody-biotin construct was purified by sequential filtration through three 7 kDa MWCO desalting columns.

The biotin conjugated anti-GFP nanobodies were mixed in a 1–1 ratio with the streptavidin coated quantum dots and incubated for 10 min at room temperature. The solution was then diluted into 1 ml of live cell medium. The cells were then incubated at room temperature with this solution for 15 min. Afterwards the supernatant was removed and replaced with regular live cell-medium (Thermo Fischer).

## SiR actin staining

The CV-1 cells were washed in 37°C warm Medium (DMEM without phenol red +FBS + glutamax). Oneμ l of the original SiR Actin (Spirochrome, 100 nM) solution was diluted in 1 ml of 37°C warm Medium (DMEM without phenol red +10% FBS +1% glutamax). To this solution, 1μ l of verapamil was added. The sample medium was removed and exchanged with this solution. After 1 hr of incubation in the incubator, the sample was imaged in a live cell imaging solution (Invitrogen) for the following hours at room temperature.

## Nucleic staining

T98G cells were washed in 37 ° C prewarmed medium (DMEM without phenol red +10% FBS +1% glutamax). For staining, 1 μl of SYTO Red Fluorescent Nucleic Acid Stain (Invitrogen) was diluted in 1 ml of complete medium and incubated with the sample for 30 min at 37 ° C in the incubator, then imaged over prolonged time.

## (*d*)STORM imaging

### Imaging buffer

Imaging buffer was made of 150 mM Tris-HCl, 1.5% β- Mercaptoethanol, 0.5% (v/w) glucose, 0.25 mg/ml glucose oxidase and 20 μg/ml catalase (pH 8.8).

### Image acquisition

The 30,000 frames of the (*d*)STORM data set have been acquired with ImSwitch on the full camera chip (1456x1,088 px). The exposure time was 20ms, the gain was set to 20 (maximal value), the black-level and offset to 0 and no time delay was set between two frames. Laser power was set to 1024 (maximal value). Setup was positioned on a regular table with a layer of foam material to dump vibrations. The room was left during the measurement. The images were acquired within around 15 min and saved directly on the computer disc in a.hdf5 file format for time efficiency and data handling advantages.

### Analysis and reconstruction:

The (d)STORM data-set was imported as a whole into ImageJ as a hdf5 file. Acquiring the full CMOS chip 30,000 times generated enormous datasets (95.6 Gb). To load the file, a Fiji plugin to load the stack as virtual stack was used. (N5, *Saalfeld et al., 2022*). The stack of images was then cropped

to retain the region of interest, which has been positioned to match the region where sufficient laser density induces blinking of single molecules during the acquisition. The analysis was done using ThunderSTORM (*Ovesný et al., 2014*), a publicly available SMLM reconstruction plugin on ImageJ. The camera parameters mentioned in the image acquisition section were used for the analysis. Image filtering was done with a Difference-of-Gaussian filter $\sigma_1 = 1.1$ px; $\sigma_2 = 1.7$ px. The approximate localization of molecules was done using the local maximum method with default settings (std(Wave.F1) as peak intensity threshold and 8-neighborhood connectivity). Sub-pixel localization of molecules was done with the PSF: Integrated Gaussian (Fitting radius 3 px) and maximum likelihood fitting method (initial sigma 1.7 px). Reconstruction of localized the raw-data was visualized using the average shifted histograms method (Magnification: 25 x; lateral shifts: 3).

## Acknowledgements

This work was funded by the Deutsche Forschungsgemeinschaft (DFG, German Research Foundation) as part of TRR 186 (Project Number 278001972) and by Freie Universität Berlin. The authors would like to thank all members of the Ewers laboratory for helpful discussions. We thank Haoran Wang for his technical support and helpful insights. We also thank the free state of Thuringia to support his project.

## Additional information

### Competing interests

Benedict Diederich: BD is a co-founder of openUC" GmbH, a commercial company that builds, delivers, supports, and integrates low-cost microscopy solutions for academic and professional use cases. The other authors declare that no competing interests exist.

### Funding

| Funder | Grant reference number | Author |
| --- | --- | --- |
| Deutsche Forschungsgemeinschaft | TRR 186 Project # 278001972 | Ando Christian Zehrer Ana Martin-Villalba Helge Ewers |

The funders had no role in study design, data collection and interpretation, or the decision to submit the work for publication.

### Author contributions

Ando Christian Zehrer, Conceptualization, Investigation, Methodology, Writing - original draft, Writing – review and editing; Ana Martin-Villalba, Resources; Benedict Diederich, Conceptualization, Resources, Software, Writing – review and editing; Helge Ewers, Conceptualization, Resources, Supervision, Funding acquisition, Writing - original draft, Project administration, Writing – review and editing

### Author ORCIDs

Ando Christian Zehrer http://orcid.org/0009-0007-4898-2727
Ana Martin-Villalba http://orcid.org/0000-0002-9405-8910
Benedict Diederich http://orcid.org/0000-0003-0453-6286
Helge Ewers http://orcid.org/0000-0003-3948-4332

Reviewer #1 (Public Review): https://doi.org/10.7554/eLife.89826.3.sa1
Reviewer #2 (Public Review): https://doi.org/10.7554/eLife.89826.3.sa2
Author Response https://doi.org/10.7554/eLife.89826.3.sa3

## Additional files

### Supplementary files
• MDAR checklist

## Data availability

All data required to reproduce the work is available under: https://github.com/openUC2/UC2-STORM-and-Fluorescence (copy archived at *openUC2, 2024b*).

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
