## [Editor Report · eLife assessment]

This **important** study provides **compelling** evidence that the low-cost and open-hardware UC2 microscopy framework can be expanded to enable single-molecule localization microscopy. The authors managed to fit the instrumentation and control thereof in a unit that can be placed in a small stage-top-incubator. Together with providing adapted software for data acquisition and data analysis, the UC.STORM setup can rival the capabilities of comparable commercial instruments at a fraction of the costs.

---

## [Referee Report · Reviewer #1 (Public Review)]

The authors have developed an open-source high-resolution microscope that is easily accessible to scientists, students, and the general public. The microscope is specifically designed to work with incubators and can image cells in culture over long periods. The authors provide detailed instructions for building the microscope and the necessary software to run it using off-the-shelf components. The system has great potential for studying cell biology and various biological processes.

The authors' work will make scientific instruments more accessible and remove obstacles to the free diffusion of capabilities and know-how in science. This important contribution will enable more people to conduct scientific research.

---

## [Referee Report · Reviewer #2 (Public Review)]

Making state-of-the-art (super-resolution) microscopy widely available has been the subject of many publications in recent years as correctly referenced in the manuscript. By advocating the ideas of open-microscopy and trying to replace expensive, scientific-grade components such as lasers, cameras, objectives, and stages with cost-effective alternatives, interested researchers nowadays have a number of different frameworks to choose from. In the iteration of the theme presented here, the authors used the existing modular UC2 framework, which consists of 3D printable building blocks, and combined a cheapish laser, detector and x,y,(z) stage with expensive filters/dichroics and an expensive high-end objective (>15k Euros).

The choice of using the UC2 framework has the advantage, that the individual building blocks can be 3D printed, although it should be mentioned that the authors used injection-moulded blocks that will have a limited availability if not offered commercially by a third party. The strength of the manuscript is the tight integration of the hardware and the software (namely the implementations of imSwitch as a GUI to control data acquisition, OS SMLM algorithms for fast sub-pixel localisation and access to Napari).

The presented experimental data is convincing, demonstrating (1) extended live cell imaging both using bright-field and fluorescence in the incubator, (2) single-particle tracking of quantum dots, and (3) and STORM measurements in cells stained against tubulin.

For the revised (current) version of the manuscript, the authors further polished the manuscript and, more importantly, added plenty of information on the GitHub page that should make it significantly easier for interested researchers to replicate the instrument.

Overall, this is compelling work that is helping to make super-resolved microscopy more accessible.

---

## [Author Response]

The following is the authors’ response to the original reviews.

**Reviewer #2 (Public Review):**
Making state-of-the-art (super-resolution) microscopy widely available has been the subject of many publications in recent years as correctly referenced in the manuscript. By advocating the ideas of open-microscopy and trying to replace expensive, scientific-grade components such as lasers, cameras, objectives, and stages with cost-effective alternatives, interested researchers nowadays have a number of different frameworks to choose from. In the iteration of the theme presented here, the authors used the existing modular UC2 framework, which consists of 3D printable building blocks, and combined a cheapish laser, detector and x,y,(z) stage with expensive filters/dichroics and a very expensive high-end objective (>15k Euros). This particular choice raises a first technical question, to which extent a standard NA 1.3 oil immersion objective available for <1k would compare to the chosen NA 1.49 one.

Measurement of the illumination quality (e.g. the spectral purity) of low budget lasers convinced us of the necessity to use spectral filtering. These cannot be replaced with lower budget alternatives, to sill retain the necessary sensitivity to image single molecules. As expected, the high-quality objectives are able to produce high-quality data. Lower budget alternatives (<500 €) to replace the objective have been tried out. Image quality is reduced but key features in fluorescent images can be identified (see figure S1). The usage of a low budget objective for SMLM imaging is possible, but quality benchmarks such as identifying railroad tracks along microtubule profiles is not possible. Their usage is not optimal for applications aiming to visualize single molecules and might find better application in teaching projects.

The choice of using the UC2 framework has the advantage, that the individual building blocks can be 3D printed, although it should be mentioned that the authors used injection-molded blocks that will have a limited availability if not offered commercially by a third party. The strength of the manuscript is the tight integration of the hardware and the software (namely the implementations of imSwitch as a GUI to control data acquisition, OS SMLM algorithms for fast sub-pixel localisation and access to Napari).

The injection-molded cubes can be acquired through the OpenUC2 platform. Alternatively, the 3D printable version of the cubes is freely available and just requires the user to have a 3D printer.https://github.com/openUC2/UC2-GIT/tree/master/CAD/CUBE_EmptyTemplate

The presented experimental data is convincing, demonstrating (1) extended live cell imaging both using bright-field and fluorescence in the incubator, (2) single-particle tracking of quantum dots, and (3) and STORM measurements in cells stained against tubulin. In the following I will raise two aspects that currently limit the clarity and the potential impact of the manuscript.First, the manuscript would benefit from further refinement. Elements in Figure 1d/e are not described properly. Figure 2c is not described in the caption. GPI-GFP is not introduced. MMS (moment scaling spectrum) could benefit from a one sentence description of what it actually is. In Figure 6, the size of the STORM and wide-field field of views are vastly different, the distances between the peaks on the tubuli are given in micrometers rather than nanometers. (more in the section on recommendations for the author)Second, and this is the main criticism at this point, is that although all the information and data is openly available, it seems very difficult to actually build the setup due to a lack of proper documentation (as of early July 2023).1. The bill of materials (https://github.com/openUC2/UC2-STORM-and-Fluorescence#bill-of-material) should provide a link to the commercially available items. Some items are named in German. Maybe split the BoM in commercially available and 3D printable parts (I first missed the option to scroll horizontally).1. The links to the XY and Z stage refer to the general overview site of the UC2 project (https://github.com/openUC2/) requiring a deep dive to find the actual information.1. Detailed building instructions are unfortunately missing. How to assemble the cubes (pCad files showing exploded views, for example)? Trouble shooting?1. Some of the hardware details (e.g. which laser was being used, lenses, etc) should be mentioned in the manuscript (or SI)I fully understand that providing such level of detail is very time consuming, but I hope that the authors will be able to address these shortcomings.

1. The bill of materials has been and will also in future still be improved. The items have been sorted into UC2 printed parts and externally acquired parts. The combination of part name as well as provider enables users to find and acquire the same parts. Additionally, depending on the country where the user is located, different providers of a given part might be advantageous as delivery means and costs might vary.

2. The Z-stage now has a specific repository with different solutions, offering different solutions with different levels of movement precision. According to the user and their budget, different solutions can be optimal for the endeavor.

https://github.com/openUC2/UC2-Zstage

The XY stage now also has a detailed repository, as the motorizing of the stage requires a fair amount of tinkering. The video tutorials and the detailed instructions on stage motorizing should help any user to reproduce the stage shown within this manuscript.https://github.com/openUC2/UC2-Motorized-XY-Table

1. The updated repository has a short video showing the general assembly of the cubes and the layers. Additionally, figure S2 shows all the pieces that are included in every layer (as a photograph as well as CAD). An exploded view of the complete setup would certainly be a helpful visualization of the complete setup. We however hope that the presented assembly tutorials and documents are sufficient to successfully reproduce the U.C.STORM setup.

First, we want to thank the reviewers for their effort to help us improving our work. We apologize for any trivial mistakes we had overlooked. Please find below our answers to the very constructive and helpful comments of the editors.

**Recommendations for the authors:**

**Reviewer #1 (Recommendations for The Authors):**
To complement the current data set:Figure 2(a & b): Panels i & ii, were chosen on the area where the distribution of the laser appears to be flatter. Can the authors select microtubules from a different section? Otherwise, it is reasonable to also crop the field-of-view along the flatter area (as done in Fig 6).

Figure 2 was changed to according to the reviewer’s suggestions. The profiles of microtubules from a different section have similar profiles, but the region with best illumination thus best SNR of the profile have been used for the figure.

Figure 2(c): The current plot shows the gaussian distribution which does not appear to be centered. Instead of a horizontal line, can the authors provide a diagonal profile across the field of view and update the panel below?

A diagonal cross-section of the illuminated FOV is provided in figure 2 to replace the previous horizontal profile. The pattern seems not to be perfectly radially symmetric, and more light seems to be blocked at the bottom of the illumination pattern compared to the top. A possible improvement can be provided by a fiber-coupled laser, that could provide a more homogeneous illumination while being easier to handle in the assembly process.

**Author response image 1. sa3fig1:** Diagonal cross-section of the illuminated FOV. Pixel-size (104nm) is the same as in figure 2. Intensity has been normalized according to the maximal value.

Figure 2(d): The system presents a XY drift of ~500nm over the course of a couple of hours. However, is not clear how the focus is being maintained. Can the authors clarify this point and add the axial drift to the plot?

The axial position of the sample could be maintained over a prolonged period of time without correcting for drift. Measurements where an axial shift was induced by tension pulses in the electronics have been discarded, but the stability of the stage seems to be sufficient to allow for imaging without lateral and axial drift correction. The XY drift measurement displayed in Figure 2(d) can be extended by measuring the σ of the PSF over time. The increase of σ would suggest an axial displacement in relation to the focus plane. In these measurements, a slight axial drift can be seen, the fluorescent beads however can still be localized over the whole course of the measurement.

A separate experiment was performed, using the same objective on the UC2 setup and on a high-quality setup equipped with a piezo actuator able to move in 10 nm steps. The precise Z steps of the piezo allows to reproducibly swipe through the PSF shape and to give an estimate of the axial displacement of the sample, according to the changes in PSF FWHM (Full Width at Half Maximum). When superimposing the graph with the UC2 measurement of fluorescent beads with the smallest possible Z step, an estimate about the relative axial position of the sample can be provided. The accuracy of the stage however remains limited.

**Author response image 2. sa3fig2:** Drift Figure: a. Drift of fluorescent TS beads on the UC2 setup positioned upon an optical table over a duration of two hours. Beads are localized and resulting displacement in i. and ii. are plotted in the graphs below. The procedure is repeated in b. with the microscope placed on a laboratory bench instead. c. (for the optical table i.) and d. (for the laboratory bench i.) show the variation in the sigma value of the localized beads over the measurement duration. As the sigma values changes when the beads are out of focus, the stability of the setup can be confirmed, as it remains practically unchanged over the measurement duration.

**Author response image 3. sa3fig3:** Z-focus Figure: Estimation of the axial position of TS beads on the UC2 setup. a. The change in PSF FWHM was quantified by acquiring a Z stack of a beads sample. The homebuilt high-quality setup (HQ) was used as a reference, by using the same objective and TS sample. The PSF FWHM on the UC2 setup was measured using the lowest possible axial stage displacement. A Z-position can thus be estimated for single molecules, as displayed in b.

Addressing the seemingly correlated behavior of the X and Y drift:

Further measurement show less correlation between drift in X and in Y. Simultaneous motion in X and Y seems to indicate that the stage or the sample is tilted. The collective movement in X and Y seems accentuated by bigger jumps, probably originating from vibrations (as more predominantly shown in the measurements on the laboratory bench compared to the optical table). Tension fluctuations inducing motion of the stage are possible but are highly unlikely to have induced the drift in the displayed measurements.

Figure 3: Can the authors comment on the effect or otherwise potential effect of the incubator (humidity, condensation etc) may have on the system (e.g., camera, electronics etc)?

When moving the microscope into the incubator, the first precaution is to check if the used electronics are able to perform at 37° C. Then, placing the microscope inside the incubator can induce condensation of water droplets at the cold interfaces, potentially damaging the electronics or reducing imaging quality. This can be prevented by preheating the microscope in e.g. an incubator without humidity, for a few hours before placing it within the functional incubator. The used incubator should also be checked for air streams (to distribute the CO2), and a direct exposure of the setup to the air stream should be prevented. The usage of a layer of foam material (e.g. Polyurethane) under the microscope helps to reduce possible effects of incubator vibrations on the microscope. The hydrophilic character of PLA makes its usage within the incubator challenging due to its reduced thermal stability. The temperature also inherently reduces the mechanical stability of 3D printed parts. Using a less hydrophilic and more thermally stable plastic, such as ABS, combined with a higher percentage of infill are the empirical solution to this challenge. Further options and designs to improve the usage of the microscope within the incubator are still in developement.

Figure 5: Can the authors perform single molecule experiments with an alternative tag suchas Alexa647?

The SPT experiments were performed with QDs to make use of their photostability and brightness. The dSTORM experiment suggests that imaging single AF647 molecules with sufficient SNR is possible. The usage of AF647 for SPT is possible but would reduce the accuracy of the localization and shorten the acquired track-lengths, due to the blinking properties of AF647 when illuminated. The tracking experiment with the QDs thus was a proof of concept that the SPT experiments are possible and allow to reproduce the diffusion coefficients published in common literature. The usage of alternative tags can be an interesting extension of the capabilities that users can perform for their applications.

Figure 6: The authors demonstrate dSTORM of microtubules. It would enhance the paper to also demonstrate 3D imaging (e.g., via cylindrical lens).

The usage of a cylindrical lens for 3D imaging was not performed yet. The implementation would not be difficult, given the high modularity of the setup in general. The calibration of the PSF shape with astigmatism might however be challenging as the vertical scanning of the Z-stage lacks reliability in its current build. Methods such as biplane imaging might also be difficult to implement, as the halved number of photons in each channel leads to losses in the accuracy of localization. As a future improvement of the setup, the option of providing 3D information with single molecule accuracy is definitely desirable and will be tried out. In the following figure, two concepts for introducing 3D imaging capabilities in the detection layer of the microscope are presented.

**Author response image 4. sa3fig4:** 3D concept Figure: Two possible setup modifications to provide axial information when imaging single molecules. a. A cylindrical lens can be placed to induce an asymmetry between the PSF FWHM in x and in y. Every Z position can be identified by two distinct PSF FWHM values in X and Y. b. By splitting the beam in two and defocusing one path, every PSF will have a specific set of values for its FWHM on the two detectors.

Imaging modalities section: Regarding the use of cling film to diffuse; can the authors comment on the continual use of this approach, including its degradation over time?

The cling foil was only used as a diffuser for broadening the laser profile. A detailed analysis of the constitution of the foil was not done, as no visible changes could be seen on the illumination pattern and the foil itself. The piece of cling foil is attached to a rotor. Detaching of the cling foil or vibrations originating from the rotor need to be minimized. By keeping the rotation speed to a necessary minimum and attaching the cling foil correctly to the rotor, a usable solution can be created. The low price of the cling foil provides the possibility to exchange the foil on a regular basis, allowing to keep the foil under optimal conditions.

**Author response image 5. sa3fig5:** Profile Figure: By moving a combination of pinhole and photometer to scan through the laser profile with a translational mount, the shape of the laser beam can be estimated. The cling foil plays the same role as a diffuser in other setups.

**Reviewer #2 (Recommendations for The Authors):**
lines20, add "," after parts110, rotating cling foil?112/116, "custom 3D printed" I thought they were injection molded, please finalize113, "puzzle pieces" rephrase and they are also barely visible119, not clear that the stage is a manual stage that was turned into a motorised one by adding belts123-126, detail for SI,132, replace Arduino-coded with Arduino-based143, add reference to Napari146, (black) cardboard seems to be a cheaper and quicker alternative153, dichroic151-155, reads more like a blog post than a paper (maybe add a section on trouble shooting)156, antibody?167/189, moderate, please be specific194, layer of foam material, specify221, add description/reference to GPI. What is that? why is it relevant?226: add one sentence description of MMS318, add "," after students332-334, as mentioned earlier, not clear, you bought a manual stage and connected belts, correct?376-377, might be difficult to understand for the layman391, what laser was used?Figure 1, poor contrast between components, components visible should be named as much as possible, maybe provide the base layer in a different shade. To me, the red and blue labels look like fluorophores.Figure 1. looks like d is the excitation layer and not e, please fix.Figure 2, caption a-c, figure 1-d!, btw, why is the drift so anti-correlated?Figure 6 (line 259) nanometer I guess, not micrometer

We now incorporated all the above-mentioned changes in the manuscript. Furthermore we added the supplementary Figures as below.

**Author response image 6. sa3fig6:** Basic concept of the UC2 setup: Left: Cubes (green) are connected to one another via puzzle pieces (white). Middle: 3D printed mounts have been designed to adapt various optics (right) to the cube framework. Combined usage of cubes and design of various mounts allows to interface various optics for the assembly.

**Author response image 7. sa3fig7:** Building the UC2 widefield microscope: a. Photograph of the complete setup. b. All pieces necessary to build the setup. A list of the components can be found in the bill of materials. c. Bottom emission layer of the microscope before assembly. d. Emission layer after assembly. Connection between cubes is doubled by using a layer of puzzles on the top and the bottom of the emission layer. e. CAD schematic of the emission layer and the positioning of the optics. f. Middle excitation layer of the microscope before assembly. Beam magnifier and homogenizer have been left out for clarity. g. Excitation layer after assembly is also covered by a puzzle layer. h. CAD schematic of the excitation layer and the positioning of the optics. i. Z-stage photograph and corresponding CAD file. Motor of the stage is embedded within the bottom cube. j. A layer of empty cubes supports the microscope stage. k. At this stage of the assembly, the objective is screwed into the objective holder. l. Finally, the stage is wired to the electronics and can then be mounted on top of the microscope (see a.).

**Author response image 8. sa3fig8:** Measurements performed on the UC2 setup with lower budget objectives. The imaged sample is HeLa cells, stably transfected to express CLC-GFP, then labelled with AF647 through immunostaining. The setup has been kept identical except for the objectives. Scale bar respectively represents 30 µm.